# Automatic Dataset Construction (ADC): Sample Collection, Data Curation, and Beyond

## Abstract

Large-scale data collection is essential for developing personalized training data, mitigating the shortage of training data, and fine-tuning specialized models. However, creating high-quality datasets quickly and accurately remains a challenge due to annotation errors, the substantial time and costs associated with human labor. To address these issues, we propose Automatic Dataset Construction (ADC), an innovative methodology that automates dataset creation with negligible cost and high efficiency. Taking the image classification task as a starting point, ADC leverages LLMs for the detailed class design and code generation to collect relevant samples via search engines, significantly reducing the need for manual annotation and speeding up the data generation process. Despite these advantages, ADC also encounters real-world challenges such as label errors (label noise) and imbalanced data distributions (label bias). We provide open-source software that incorporates existing methods for label error detection, robust learning under noisy and biased data, ensuring a higher-quality training data and more robust model training procedure. Furthermore, we design three benchmark datasets focused on label noise detection, label noise learning, and class-imbalanced learning. These datasets are vital because there are few existing datasets specifically for label noise detection, despite its importance. Finally, we evaluate the performance of existing popular methods on these datasets, thereby facilitating further research in the field.

## 1 Introduction

In the era of Large Language Models (LLMs), the literature has observed an escalating demand for fine-tuning specialized models (Benary et al., 2023; Porsdam Mann et al., 2023; Woźniak et al., 2024), highlighting the urgent need for customized datasets (Wu et al., 2023; Lyu et al., 2023; Tan et al., 2024).

**Traditional Dataset Construction (TDC)** typically involves sample collection followed by labor-intensive annotation, requiring significant human efforts (Xiao et al., 2015; Krizhevsky et al., 2009; Wei et al., 2021; Liu et al., 2015). Consequently, TDC is often hindered by the limitations of human expertise, leading to suboptimal design (Ramaswamy et al., 2023), data inaccuracies (Natarajan et al., 2013; Liu & Tao, 2015; Li et al., 2017; Xiao et al., 2015; Wei et al., 2022b), and extensive manual labor (Chang et al., 2017; Kulesza et al., 2014). Furthermore, certain datasets are inherently challenging or risky to collect manually, such as those for fall detection in elderly individuals, dangerous activities like extreme sports, and network intrusion detection. Therefore, there is a growing need for more automated and efficient data collection methods to enhance accuracy and efficiency in dataset creation (Bansal et al., 2021b;a; Han et al., 2021). To address these challenges, we propose the **Automatic Dataset Construction (ADC)**, an innovative approach designed to construct customized large-scale datasets with minimal human involvement. Our methodology reverses the traditional process by starting with detailed annotations that guide sample collection. This significantly reduces the workload, time, and cost associated with human annotation, making the process more efficient and targeted for LLM applications, ultimately outperforming traditional methods.

**Traditional-Dataset-Construction v.s. Automatic Dataset Construction** Figure 1 illustrates the difference between Traditional Dataset Construction (TDC) and Automatic Dataset Construction (ADC). TDC typically unfolds in two stages: developing classification categories and employing

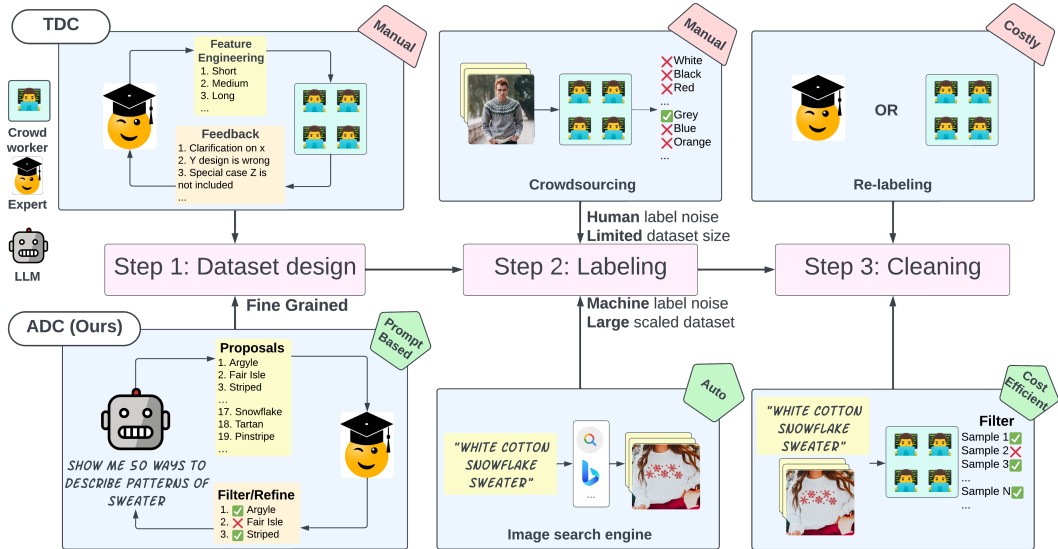

Figure 1: **Comparisons of key steps in dataset construction**. In **Step 1**: Dataset design, ADC utilizes LLMs to search the field and provide instant feedback, unlike traditional methods that rely on manual creation of class names and refine through crowdsourced worker feedback. In **Step 2**: Labeling, ADC reduces human workload by flipping the data collection process, using targets to search for samples. In **Step 3**: Cleaning, ADC instructs human labor to filter noisy labeled samples from previous steps, instead of relabeling.

human labor for annotation. Creating comprehensive categories requires deep domain knowledge and experience, tasks that even expert researchers find challenging (Ramaswamy et al., 2023). Crowdsourcing is often used to refine these categories, but it increases time and costs without necessarily improving label quality (Chang et al., 2017; Kulesza et al., 2014). Annotation by human workers introduces label noise, which impacts dataset reliability, even when multiple inputs are aggregated (Sheng et al., 2008). In contrast, ADC offers improvements at each key step. In the "Dataset design", ADC uses LLMs to automate field searches and provide instant feedback, unlike traditional manual class and attribute creation. In the sample annotation steps, ADC reverses the labeling process by using predefined targets to search for samples, human annotators are then instructed to filter noisy labeled samples, significantly reducing the need for costly human annotation.

Our main contributions can be summarized as follows:

- **The Automatic-Dataset-Construction (ADC) Pipeline:** We introduce Automatic-Dataset-Construction (ADC), an automatic data collection pipeline that requires minimal human efforts, tailored for the specialized large-scale data collection. The code of ADC pipeline will be released after accepted, which easily adaptable to any image-related high-quality dataset construction.

- **Software Efforts for Addressing Dataset Construction Challenges:** We explore several challenges observed in real-world dataset construction, including detecting label errors, learning with noisy labels, and class-imbalanced learning. To improve the quality of the constructed data and model training, we provide well-written software that incorporates existing solutions to these challenges. Data curation code will be released after paper acceptance.

- **Dataset and Benchmark Efforts:** Leveraging ADC, we developed Clothing-ADC, an image dataset containing one million images with over 1,000 subclasses for each clothing type. Our dataset offers a rich hierarchy of categories, creating well-defined sub-populations that support research on a variety of complex and novel tasks. To further facilitate the exploration of the aforementioned challenges (label noise detection and learning, class-imbalanced learning), we customize three benchmark subsets and provide benchmark performances of the implemented methods in our software. This offers researchers a platform for performance comparisons, enhancing the evaluation and refinement of their approaches.

## 2 AUTOMATIC-DATASET-CONSTRUCTION (ADC)

Traditional methods are invaluable for discovering new knowledge, particularly in fields like citizen science. The efforts of experts in these domains are irreplaceable, and we respect the dedication required to collect and annotate data in these contexts. However, collecting a dataset from the traditional pipeline requires tens of thousand of human labor hours to annotate each sample (Van Horn et al., 2018; Deng et al., 2009). Despite the high effort from human experts, obtaining a clean dataset is very hard under traditional collection methods (Northcutt et al., 2021b).

Our proposed ADC pipeline serves a different purpose. Rather than attempting to replace human experts by synthetic labels from models, our ADC provides assistance in collecting existing data from the internet. In this section, we discuss the detailed procedure of ADC, as well as an empirical application.

### 2.1 THE ADC PIPELINE

The ADC pipeline generates datasets with finely-grained class and attribute labels, utilizing data diagnostic software to perform data curation. Below, we provide a step-by-step guide to collecting the Clothing-ADC, a clothes image dataset, along with an overview of its statistics and key information. The overall Automatic-Dataset-Construction (ADC) pipeline is illustrated in Figure 1.

**Step 1: Dataset design with large language models (LLM)**

- **Detailed Categories Identification:** LLMs assist researchers in conducting a more thorough search in the field by processing and analyzing numerous concepts simultaneously, unlike humans who may overlook certain factors when faced with a large volume of concepts (Ramaswamy et al., 2023). We utilize LLMs to identify attribute types for each class. Then use a prompt of *"Show me <30-80> ways to describe <Attribute> of <Class>"* to generate the proposed subclasses.

- **Iterative Refinement:** The initial category list generated by the LLM undergoes review and refinement either by domain experts or through self-examination by the LLM itself, ensuring alignment with specific application or research needs, as shown in Figure 2. This iterative refinement process enables the creation of a high-quality dataset with finely-grained class labels. Additionally, this approach facilitates rapid iterative feedback during the design phase, offering a significant advantage over traditional methods that rely on annotator feedback during the test run annotation phase. This acceleration enables researchers to explore and refine their ideas more efficiently, resulting in better dataset quality and reduced development time.

- **LLM Hallucination Issues:** LLMs and VLMs are capable of providing synthetic sample labeling. However, they tend to hallucinate excessively and uncontrollably (Xu et al., 2024; Huang et al., 2023). To create a responsible dataset, we limit the use of LLMs to the dataset design phase, where they assist human designers. Any hallucinated or inaccurate labels should be caught.

> Help me pick the best color options to describe sweaters from the many choices I've **found**:
> Ash, black, blue ... Green, Grey ... Indigo, Ivory ... <More attribute designs>
> **First** remove the **wrongly labeled** and **repetitive** samples. Then **organize** them by categories.

> [LLM]: **I've removed the duplicates and categorized the colors for you:**
> *Neutrals*: Ash, Beige, Black, Camel, Charcoal, Dark Gray, ...
> ... <More categories>

> Very Good. I would like to have a diverse range of color, help me to get a set of 10 from this list.

> [LLM]: **I've selected a diverse set of 10 colors for you:**
> Navy (*Blues*) Forest Green (*Greens*) Burgundy (*Reds*) Mustard (*Yellows*) Charcoal (*Neutrals*)
> Teal (*Greens*) Plum (*Purples*) Light Blue (*Blues*) Tan (*Earth* Tones) Coral (*Pinks*)

Figure 2: Example of using LLM for iterative refinement of attribute designs.

**Step 2: Automated Labeling**   For image data collection and labeling, ADC utilizes APIs provided by Google or Bing for automated querying, guaranteeing real samples are collected from the web. Each category and attribute identified in the first step can be used to formulate search queries, which is the sample label also.

**Step 3: Data Curation and Cleaning**

- **Algorithmic Label Noise Detection:** For applications where some label noise can be tolerated, existing data curation software capable of identifying and filtering out irrelevant images, such as Docta, CleanLab , and Snorkel [1], etc. For example, these tools can identify when an item is mislabeled regarding its type, material, or color. Finally, ADC aggregates the suggested labels recommended by the dataset curation software and removes potentially mislabeled or uncertain samples. For illustration, we adopt a data-centric label curation software (Docta) in Algorithm 1. The high-level idea of this algorithm is to estimate the essential label noise transition matrix $T\_Est$ without using ground truth labels, achieved through the consensus equations (**Part A**). Following this, Algorithm 1 identifies those corrupted instances via the cosine similarity ranking score among features as well as a well-tailored threshold based on the obtained information (i.e., $T\_Est$), and then relabels these instances using KNN-based methods (**Part B**). For more details, please refer to work (Zhu et al., 2023; 2021; 2022).

---

**Algorithm 1** Data centric curation (Docta)

---

1: **procedure** DOCTA(noisyDataset, preTrainedModel)
2:     **Part A:** Encode images and estimate label noise transition matrix
3:     $features \leftarrow$ EncodeImages(noisyDataset, preTrainedModel)
4:     $T\_Est \leftarrow$ EstimateTransitionMatrix($features, noisyLabels$)
5:     **Part B:** Identify and relabel corrupted instances
6:     $corruptedInstances \leftarrow$ SimiFeat-rank($features, noisyLabels, T\_Est$)
7:     $curedLabels \leftarrow$ KNN-based Relabeling($corruptedInstances$)
8:     **Return** $curedLabels$
9: **end procedure**

---

- **Cost Efficient Human-in-the-Loop:** For domains requiring clean data, we advocate for human involvement in addition to algorithmic approaches to ensure perfect annotations. Unlike traditional pipelines where humans are asked to relabel samples from scratch, our ADC pipeline provides a large amount of noisy labeled samples for humans to review and select the accurate ones. This approach is mentally easier and results in a clean dataset, as the selected samples have guaranteed human and machine label agreements. Analyses of human votes are in Appendix B.

## 2.2 CLOTHING-ADC

To illustrate the ADC pipeline, we present the Clothing-ADC dataset, which comprises a substantial collection of clothing images. The dataset includes 1,076,738 samples, with 20,000 allocated for evaluation, another 20,000 for testing, and the remaining samples used for training. Each image is provided at a resolution of 256x256 pixels. The dataset is categorized into 12 primary classes, encompassing a total of 12,000 subclasses, with an average of 89.73 samples per subclass. Detailed statistics of the dataset are provided in Table 1. The following subsection elaborates on the dataset construction process in comprehensive detail. Other ADC application examples are in Appendix D.

**Subclass Design**     Utilizing GPT-4, we identified numerous attribute options for each clothing type. For example, in the case of sweaters, we recognized eight distinct attributes: color, material, pattern, texture, length, neckline, sleeve length, and fit type. The language model was able to find 30-50 options under each attribute. Our Clothing-ADC dataset includes the three most common attributes: color, material, and pattern, with each attribute having ten selected options. This results in 1000 unique subclasses per clothing type. The selected attributes are detailed in Table 6 (Appendix).

**Data Collection**     The ADC pipeline utilizes the Google Image API to collect clothing images by formulating queries that include attributes such as "Color + Material + Pattern + Cloth Type" (e.g., "white cotton fisherman sweater"). Figure 3 shows examples of these queries and the corresponding images retrieved. The relevance of the search results tends to decline after a significant number of samples are gathered, leading us to set a cutoff threshold of 100 samples per query. After removing broken links and improperly formatted images, each subclass retained approximately 90 samples. These queries generated noisy, webly-labeled data for the training set.

---

[1]Docta:www.docta.ai, CleanLab:www.cleanlab.ai, Snorkel:www.snorkel.ai

| Dataset Overview | |
|---|---|
| Number of Samples | $1,076,738$ |
| Resolution | $256 \times 256$ |
| **Dataset Split** | |
| Train set(with web noise) | $1,036,738$ |
| Evaluation set (Clean) | 20,000 |
| Test set (Clean) | 20,000 |
| **Classification Structure** | |
| Main Class | 12 |
| Total Subclasses | 12,000 |
| **Subclass Details** | |
| Attribute (Color) | 10 |
| Attribute (Material) | 10 |
| Attribute (Pattern) | 10 |
| Ave. Samples per attribute | 89.73 |

Table 1: Dataset information summary of Clothing-ADC Dataset.

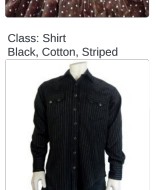
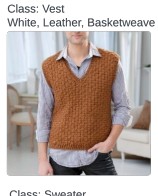
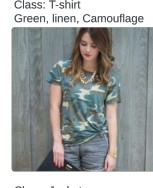
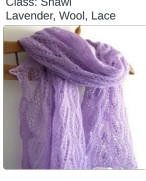
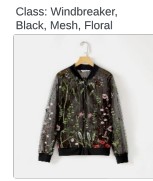

Class: Dress — Brown, Velvet, Polka dot
Class: Vest — White, Leather, Basketweave
Class: T-shirt — Green, linen, Camouflage
Class: Shawl — Lavender, Wool, Lace
Class: Shirt — Black, Cotton, Striped
Class: Sweater — Yellow, Mohair, Intarsia
Class: Jacket — Orange, Denim, Moto
Class: Windbreaker, — Black, Mesh, Floral

Figure 3: Samples from the collected Clothing-ADC Dataset

**Creating Test Set**   Note that the collected samples may suffer from web-based label noise, where annotations might be incorrect due to mismatches provided by search engines, the traditional approach typically involves manually re-labeling existing annotations and aggregating multiple human votes per label to ensure a high-quality subset for testing purposes. Our ADC pipeline enhances efficiency by presenting annotators with a set of samples that share the same machine-generated label. Annotators are then tasked with selecting a subset of correctly labeled samples, choosing a minimum of four samples out of twenty. This method significantly reduces both manual effort and difficulty, encouraging annotators to critically evaluate machine-generated labels and thereby reducing the effect of human over-trust in AI answers (Bansal et al., 2019; 2021b). The samples selected through this process are considered "clean" labels, representing a consensus between human judgment and machine-generated labels (Liu et al., 2023).

**Compare With Existing Datasets**   Table 2 provides an insightful comparison between existing datasets and Clothing-ADC. Briefly speaking, compared with existing datasets, the ADC pipeline is able to help humans without domain expertise to create fine-grained attributes for the dataset, and automatic annotation and label cleaning drastically eliminate human effort during label creation.

Table 2: Our ADC pipeline creates a large-scale image classification dataset with a clean test set. Most existing datasets require human effort for labeling, whereas our pipeline can automatically annotate and clean the data. While Clothing-ADC provides fine-grained attribute labels, our dataset design does not require human expertise in the field.

| Dataset | # Train/Test | # Classes | Noise Rate(%) | Has Attributes | Auto annotation | Require expert? |
|---|---|---|---|---|---|---|
| **iNaturalist** (Van Horn et al., 2018) | 579k/279k | 54k | Close to 0 | ✗ | ✗ | ✓ |
| **WebVision** (Li et al., 2017) | 2.4M/100k | 1000 | 20 | ✗ | ✓ | ✓ |
| **ANIMAL-10N** (Song et al., 2019) | 50k/10k | 10 | 8 | ✗ | ✗ | ✗ |
| **CIFAR-10N** (Wei et al., 2021) | 50k/10k | 10 | 9.03/25.60/40.21 | ✗ | ✗ | ✗ |
| **CIFAR-100N** (Wei et al., 2021) | 50k/10k | 100 | 25.6/40.2 | ✗ | ✗ | ✗ |
| **Food-101N** (Bossard et al., 2014) | 75.75k/25.25k | 101 | 18.4 | ✗ | ✗ | ✓ |
| **Clothing1M** (Xiao et al., 2015) | 1M in all | 14 | 38.5 | ✗ | ✗ | ✓ |
| **Clothing-ADC (Ours)** | 1M/20k | 12 | 22.2-32.7 | 12k | ✓ | ✗ |

## 3 CHALLENGE ONE: DEALING WITH IMPERFECT DATA ANNOTATIONS

The first pervasive and critical challenge during the automatic dataset construction lies in the prevalence of noisy/imperfect labels. This issue is intrinsic to web-sourced data, which, although rich in diversity, often suffers from inaccuracies due to the uncurated nature of the internet. These errors manifest as mislabeled images, inconsistent tagging, and misclassified attributes, introducing non-negligible noise into the dataset that may adversely affect the training and performance of machine learning models. The following discussion bridges the gap between imperfect data and curated data via mining and learning with label noise, to refine data quality, enhance label accuracy, and ensure the reliability of Auto-Dataset-Construction (ADC) for high-stakes AI applications.

**Formulation** Let $D := \{(x_n, y_n)\}_{n \in [N]}$ represent the training samples for a $K$-class classification task, where $[N] := \{1, 2, ..., N\}$. Suppose that these samples $\{(x_n, y_n)\}_{n \in [N]}$ are outcomes of the random variables $(X, Y) \in \mathcal{X} \times \mathcal{Y}$, drawn from the joint distribution $\mathcal{D}$. Here, $\mathcal{X}$ and $\mathcal{Y}$ denote the spaces of features and labels, respectively. However, classifiers typically access a noisily labeled training set $\widetilde{D} := \{(x_n, \tilde{y}_n)\}_{n \in [N]}$, assumed to arise from random variables $(X, \widetilde{Y}) \in \mathcal{X} \times \widetilde{\mathcal{Y}}$, drawn from the distribution $\widetilde{\mathcal{D}}$. It is common to observe instances where $y_n \neq \tilde{y}_n$ for some $n \in [N]$. The transition from clean to noisy labels is typically characterized by a noise transition matrix $T(X)$, defined as $T_{i,j}(X) := \mathbb{P}(\widetilde{Y} = j \mid Y = i, X)$ for all $i, j \in [K]$ (Natarajan et al., 2013; Liu & Tao, 2015; Patrini et al., 2017).

## 3.1 The challenge of label noise detection

While employing human annotators to clean data is effective in improving label quality, it is often prohibitively expensive and time-consuming for large datasets. A practical alternative is to enhance label accuracy automatically by first deploying algorithms to detect potential errors within the dataset and then correcting these errors through additional algorithmic processing or crowdsourcing.

### 3.1.1 Existing approaches to detect label noise

**Learning-Centric Approaches:** Learning-centric approaches often leverage the behavior of models during training to infer the presence of label errors based on how data is learned. One effective strategy is confidence-based screening, where labels of training instances are scrutinized if the model's prediction confidence falls below a certain threshold. This approach assumes that instances with low confidence scores in the late training stage are likely mislabeled (Northcutt et al., 2021a). Another innovative technique involves analyzing the gradients of the training loss w.r.t. input data. Pruthi et al. (2020) utilize gradient information to detect anomalies in label assignments, particularly focusing on instances where the gradient direction deviates significantly from the majority of instances. Researchers have also utilized the memorization effect of deep neural networks, where models tend to learn clean data first and only memorize noisy labels in the later stages of training. Techniques that track how quickly instances are learned during training can thus identify noisy labels by focusing on those learned last (Han et al., 2019; Liu et al., 2020; Xia et al., 2020).

**Data-Centric Approaches:** Data-centric methods focus on analyzing data features and relationships rather than model behavior for detection. The ranking-based detection method (Brodley & Friedl, 1999) ranks instances by the likelihood of label errors based on their alignment with model predictions. An ensemble of classifiers evaluates each instance, flagging those that consistently deviate from the majority vote as noisy. Neighborhood Cleaning Rule Laurikkala (2001) uses the $k$-nearest neighbors algorithm to check label consistency with neighbors, identifying instances whose labels conflict with the majority of their neighbors as potentially noisy. Zhu et al. (2022) propose advanced data-centric strategies for detecting label noise without training models. Their local voting method uses neighbor consensus to validate label accuracy, effectively identifying errors based on agreement within the local feature space.

### 3.1.2 Clothing-ADC in label noise detection

We prepared a subset of 20,000 samples from the Clothing-ADC dataset for the label noise detection task, including both noisy and clean labels. We collected three human annotations for each image via Amazon MTurk. Annotators were instructed to classify the labels as correct, unsure, or incorrect. Each sample received three votes. Based on these annotations, we determined the noise rate to be 22.2%-32.7%. Using majority vote aggregation implies uncertainty of the label correctness. By using a more stringent aggregation criterion, more samples are considered as noisy labeled. Under the extreme case where any doubts from any human annotator can disqualify a sample, our auto collected dataset still retains 61.3% of its samples. For a detailed distribution of human votes, see Table 9 in the Appendix.

**Benchmark Efforts** Detection performance comparisons of certain existing solutions are given in Table 3. We adopt ResNet-50 (He et al., 2016) as the backbone model to extract the feature here. For each method, we use the default hyper-parameter reported in the original papers. All methods are

Table 3: $F_1$-**Score** comparisons among several label noise detection methods on Clothing-ADC.

| Methods | CORES (Cheng et al., 2020) | CL (Northcutt et al., 2021a) | Deep $k$-NN (Papernot & McDaniel, 2018) | Simi-Feat (Zhu et al., 2022) |
|---------|---------------------------|------------------------------|------------------------------------------|------------------------------|
| $F_1$-**Score** | 0.4793 | 0.4352 | 0.3991 | 0.5721 |

tested on 20,000 points and predict whether the data point is corrupted or not. We follow Zhu et al. (2022) to apply the baseline methods to our scenario. In Table 3, the performance is measured by the $F_1$-score of the detected corrupted instances, which is the harmonic mean of the precision and recall, i.e., $F_1 = \frac{2}{\text{Precision}^{-1} + \text{Recall}^{-1}}$. Let $v_n = 1$ indicate that the $n$-th label is detected as a noisy/wrong label, and $v_n = 0$ otherwise. Then, the precision and recall of detecting noisy labels can be calculated as: $\text{Precision} = \frac{\sum_n \mathbb{1}(v_n=1, \tilde{y}_n \neq y_n)}{\sum_n \mathbb{1}(v_n=1)}$, $\quad \text{Recall} = \frac{\sum_n \mathbb{1}(v_n=1, \tilde{y}_n \neq y_n)}{\sum_n \mathbb{1}(\tilde{y}_n \neq y_n)}$.

### 3.2 THE CHALLENGE OF LEARNING WITH NOISY LABELS

Another technique is robust learning that can effectively learn from noisy datasets without being misled by incorrect labels, thus maintaining high accuracy and reliability in real-world applications.

#### 3.2.1 EXISTING APPROACHES TO LEARN WITH LABEL NOISE

In this subsection, we contribute to the literature with robust learning software; all covered methods can be mainly summarized into the following three categories: robust loss functions, robust regularization techniques, and multi-network strategies.

**Robust Loss Designs**  Loss Correction modifies the traditional loss function to address label noise by incorporating an estimated noise transition matrix, thereby recalibrating the model's training focus (Patrini et al., 2017). Loss-Weighting strategies mitigate the impact of noisy labels by assigning lower weights to likely mislabeled instances, reducing their influence on the learning process (Liu & Tao, 2015; Ren et al., 2018). Symmetric Cross-Entropy Loss balances the contributions of correctly labeled and mislabeled instances, improving the model's resilience to label discrepancies (Wang et al., 2019). Generalized Cross-Entropy Loss, derived from mean absolute error, offers enhanced robustness against outliers and label noise (Zhang & Sabuncu, 2018). Peer Loss Functions form a family of robust loss functions (Liu & Guo, 2020; Wei & Liu, 2020; Cheng et al., 2020), leveraging predictions from peer samples as regularization to adjust the loss computation, thereby increasing resistance to noise.

**Robust Regularization Techniques**  Regularization techniques are designed to constrain or modify the learning process, thereby reducing the model's sensitivity to label noise. Mixup (Zhang et al., 2017) generates synthetic training examples by linearly interpolating between pairs of samples and their labels, enhancing model generalization and smoothing label predictions. Label Smoothing (Müller et al., 2019; Lukasik et al., 2020) combats overconfidence in unreliable labels by adjusting them towards a uniform distribution. Negative Label Smoothing (Wei et al., 2022a) refines this approach by specifically adjusting the smoothing process for negative labels, preserving model confidence in high-noise environments. Early-Learning Regularization tackles the issue of early memorization of noisy labels by dynamically adjusting regularization techniques during the initial training phase (Liu et al., 2020; Xia et al., 2020).

**Multi-Network Strategies**  Employing multiple networks can enhance error detection and correction through mutual agreement and ensemble techniques. In Co-teaching, two networks concurrently train and selectively share clean data points with each other, mitigating the memorization of noisy labels (Han et al., 2018). MentorNet (Jiang et al., 2018) equips a student network with a curriculum that emphasizes samples likely to be clean, as determined by the observed dynamics of a mentor network. DivideMix leverages two networks to segregate the data into clean and noisy subsets using a mixture model, allowing for targeted training on each set to manage label noise better (Li et al., 2020).

Table 4: Experiment results of label noise learning methods on Clothing-ADC and Clothing-ADC (tiny). We report the model prediction accuracy on the held-out clean labeled test set for comparisons.

| Methods / Dataset | Clothing-ADC | Clothing-ADC (tiny) |
|---|---|---|
| **Cross-Entropy** | 74.76 | $67.72 \pm 0.40$ |
| **Backward Correction** (Patrini et al., 2017) | 77.51 | $70.49 \pm 0.06$ |
| **Forward Correction** (Patrini et al., 2017) | 78.45 | $70.60 \pm 0.14$ |
| **(Positive) LS** (Lukasik et al., 2020) | 81.94 | $70.67 \pm 0.15$ |
| **(Negative) LS** (Wei et al., 2022a) | 78.65 | $70.14 \pm 0.13$ |
| **Peer Loss** (Liu & Guo, 2020) | 78.58 | $70.92 \pm 0.17$ |
| $f$-**Div** (Wei & Liu, 2020) | 77.43 | $68.98 \pm 0.22$ |
| **Divide-Mix** (Li et al., 2020) | 77.00 | $71.58 \pm 0.11$ |
| **Jocor** (Wei et al., 2020) | 78.47 | $72.81 \pm 0.02$ |
| **Co-Teaching** (Han et al., 2018) | 80.49 | $70.55 \pm 0.08$ |
| **LogitCLIP** (Wei et al., 2023a) | 77.85 | $70.16 \pm 0.14$ |
| **TaylorCE** (Chen et al., 2022) | 81.87 | $71.11 \pm 0.07$ |

### 3.2.2 CLOTHING-ADC IN LABEL NOISE LEARNING

We provide two versions of the Label Noise Learning task, Clothing-ADC and Clothing-ADC (tiny). Specifically, Clothing-ADC leverages the whole available (noisy) training samples to construct the label noise learning task. The objective is to perform class prediction w.r.t. 12 clothes types: Sweater, Windbreaker, T-shirt, Shirt, Knitwear, Hoodie, Jacket, Suit, Shawl, Dress, Vest, Underwear. We also provide a tiny version of Clothing-ADC, which contains 50K training images, sharing similar size with certain widely-used ones, i.e., MNIST, Fashion-MNIST, CIFAR-10, CIFAR-100, etc.

**Estimated Noise Level of Clothing-ADC** We selected a subset of 20,000 training samples and asked human annotators to evaluate the correctness of the auto-annotated dataset. After aggregating three votes from annotators, we estimate the noise rate to be 22.2%-32.7%, which consists of 10.5% of the samples having ambiguity and 22.2% being wrongly labeled. The remaining 77.8% of the samples were correctly labeled. The detailed distribution of human votes is given in Appendix Table 9.

**Benchmark Efforts** In this task, we aim to provide the performance comparison among various learning-with-noisy-label solutions. All methods utilize ResNet-50 as the backbone model and are trained for 20 epochs to ensure a fair comparison. We report the model prediction accuracy on the held-out clean labeled test set. For the tiny version, we conduct three individual experiments using three different random seeds and calculate the mean and standard deviation. As shown in Table 4, certain methods, such as Positive LS and Taylor CE, significantly outperform Cross-Entropy. These results underscore the importance and necessity of pairing ADC with robust learning software.

## 4 CHALLENGE TWO: DEALING WITH IMBALANCED DATA DISTRIBUTION

We now discuss another real-world challenge: when imperfect annotations meet with imbalanced class/attribute distributions. As shown in Figure 4, long-tailed data distribution is a prevalent issue in web-based datasets: to collect a dataset of wool suits without a specified target color on Google Image, the majority would likely be dark or muted shades (grey, black, navy), with few samples in brighter colors like pink or purple. This natural disparity results in most data points belonging to a few dominant categories, while the remaining are spread across several minority groups.

We are interested in how class-imbalance intervenes with learning. In real-world scenarios, the distribution of classes tends to form a long-tail form, in other words, the head class and the tail class differ significantly in their sample sizes, i.e., $\max_k \mathbb{P}(Y = k) \gg \min_{k'} \mathbb{P}(Y = k')$.

### 4.1 EXISTING APPROACHES FOR CLASS IMBALANCE LEARNING

**Data-Level Methods** Data-level methods modify training data to balance class distribution, focusing on adjusting the dataset by increasing minority class instances or decreasing majority class instances. Oversampling increases the number of minority class instances to match or approach the majority class. This can be done through simple duplication (Jo & Japkowicz, 2004) (e.g., random oversampling) or generating synthetic data (Chawla et al., 2002; Han et al., 2005; Bunkhumpornpat et al., 2009; He et al., 2008). Undersampling reduces the number of majority class instances, helping

Figure 4: Long-tailed data distribution is a prevalent issue in many datasets. Searching "wool suit" in Google image results in dark wool suits, while only a few are of a light color (red/pink).

to balance class distributions but potentially discarding useful information (Mani & Zhang, 2003; Kubat et al., 1997; TOMEK, 1976).

**Algorithm-Level Methods**    These methods adjust the training process or model to handle unequal class distributions better. Specifically, cost-sensitive learning assigns different costs to misclassifications of different classes, imposing higher penalties for errors on the minority class (Elkan, 2001). It modifies the loss function to incorporate misclassification costs, encouraging the model to focus more on minority class errors (Kukar et al., 1998; Zhou & Liu, 2005). Thresholding adjusts the decision threshold for class probabilities to account for class imbalance. Instead of using a default threshold, different thresholds are applied based on class distribution, modifying the decision process for predicting class labels (Lawrence et al., 2002; Richard & Lippmann, 1991).

## 4.2 CLOTHING-ADC IN CLASS-IMBALANCED LEARNING

Note that in the label noise learning task, the class distributes with almost balanced prior. However, in practice, the prior distribution is often long-tail distributed. Hence, the combined influence of label noise and long-tail distribution is a new and overlooked challenge presented in the literature. To facilitate the exploration of class-imbalanced learning, we tried to reduce the impact of noisy labels via selecting high-quality annotated samples as recognized by dataset curation software. Human estimation suggested a noise rate of up to 22.2%, and 10.5% marked as uncertain. To address this, we employed two methods to remove noisy samples: a data centric curation (Algorithm 1), which removed 26.36% of the samples, and a learning-centric curation (Appendix Algorithm 3), which removed 25%. Combined, these methods eliminated 45.15% of the samples, with an overlap of 6.21% between the two approaches. We provide Clothing-ADC CLT, which could be viewed as the long-tail (class-level) distributed version of Clothing-ADC. Denote by $\rho$ the imbalanced ratio between the maximum number of samples per class and the minimum number of samples per class. In practice, we provide $\rho = 10, 50, 100$ (class-level) long-tail version of Clothing-ADC.

**Benchmark Efforts**    Regarding the evaluation metric, we follow from the recently proposed metric Wei et al. (2023b), which considers an objective that is based on the weighted sum of class-level performances on the test data, i.e., $\sum_{i \in [K]} g_i \text{Acc}_i$, where $\text{Acc}_i$ indicates the accuracy of the class $i$:

$$\delta\text{-worst accuracy:} \qquad \min_{g \in \Delta_K} \sum_{i \in [K]} g_i \text{Acc}_i, \quad \text{s.t. } D(\mathbf{g}, \mathbf{u}) \leq \delta .$$

Here, $\Delta_K$ denotes the $(K-1)$-dimensional probability simplex, where $K$ is the number of classes as previously defined. Let $\mathbf{u} \in \Delta_K$ be the uniform distribution, and $\mathbf{g} := [g_1, g_2, ..., g_K]$ is the class weights. The $\delta$-worst accuracy measures the worst-case $\mathbf{g}$-weighted performance with the weights constrained to lie within the $\delta$-radius ball around the target (uniform) distribution. For any chosen divergence $D$, it reduces to the mean accuracy when $\delta = 0$ and to the worst accuracy for $\delta \to \infty$. The objective interpolates between these two extremes for other values of $\delta$ and captures our goal of optimizing for variations around target priors instead of more conventional objectives of optimizing for either the average accuracy at the target prior or the worst-case accuracy.

Different from the previous dataset we used in noise learning, we use a cleaner dataset for this class-imbalance learning to avoid the distractions of noisy labels. The size of this dataset consists of 56,2263 images rather than 1M. The backbone model we use is ResNet-50. For the class distributions for different $\rho$, we include them in the Appendix. All the experiments are run for 5 times and we calculate the mean and standard deviation. With the imbalance ratio going larger, the accuracy becomes worse, which is expected for a more difficult task.

Table 5: $\delta$-worst accuracy of class-imbalanced learning baselines on clothing-ADC CLT dataset.

| Method | $\delta = 0$ Worst Accuracy | | | $\delta = 1$ Worst Accuracy | | | $\delta = \infty$ Worst Accuracy | | |
|---|---|---|---|---|---|---|---|---|---|
| | $\rho = 10$ | $\rho = 50$ | $\rho = 100$ | $\rho = 10$ | $\rho = 50$ | $\rho = 100$ | $\rho = 10$ | $\rho = 50$ | $\rho = 100$ |
| **Cross Entropy** | $57.80 \pm 0.25$ | $33.85 \pm 0.13$ | $30.10 \pm 0.22$ | $19.79 \pm 0.23$ | $0.35 \pm 0.11$ | $0.00 \pm 0.00$ | $0.96 \pm 0.26$ | $0.00 \pm 0.00$ | $0.00 \pm 0.00$ |
| **Focal** (Lin et al., 2017) | $72.70 \pm 0.19$ | $65.17 \pm 0.29$ | $62.28 \pm 0.31$ | $49.66 \pm 1.09$ | $34.14 \pm 1.05$ | $29.12 \pm 0.92$ | $38.12 \pm 1.76$ | $19.46 \pm 1.49$ | $13.44 \pm 1.73$ |
| **LDAM** (Cao et al., 2019) | $72.50 \pm 0.15$ | $65.70 \pm 0.26$ | $63.25 \pm 0.35$ | $51.13 \pm 0.78$ | $36.86 \pm 1.03$ | $30.88 \pm 1.07$ | $40.90 \pm 1.53$ | $23.24 \pm 1.69$ | $15.69 \pm 2.13$ |
| **Bal-Softmax** (Ren et al., 2020) | $74.18 \pm 0.08$ | $70.48 \pm 0.55$ | $69.47 \pm 0.44$ | $56.57 \pm 0.93$ | $53.37 \pm 2.31$ | $44.24 \pm 2.83$ | $48.54 \pm 2.27$ | $45.64 \pm 3.98$ | $50.60 \pm 1.40$ |
| **Logit-Adjust** (Menon et al., 2020) | $74.08 \pm 0.05$ | $70.94 \pm 0.24$ | $69.44 \pm 0.18$ | $56.00 \pm 1.39$ | $53.93 \pm 2.46$ | $49.70 \pm 2.64$ | $47.45 \pm 2.26$ | $47.76 \pm 4.07$ | $43.26 \pm 4.69$ |
| **Post-hoc** (Menon et al., 2020) | $62.54 \pm 0.11$ | $54.84 \pm 0.15$ | $49.63 \pm 0.71$ | $35.67 \pm 0.49$ | $24.14 \pm 1.18$ | $19.00 \pm 0.68$ | $22.50 \pm 0.78$ | $7.15 \pm 1.82$ | $3.81 \pm 0.97$ |
| **Drops** (Wei et al., 2023b) | $73.66 \pm 0.29$ | $69.14 \pm 0.38$ | $67.15 \pm 0.17$ | $58.12 \pm 0.26$ | $47.07 \pm 0.74$ | $43.42 \pm 1.19$ | $50.85 \pm 0.49$ | $36.27 \pm 1.15$ | $32.43 \pm 1.90$ |

## 5 LIMITATION

While our proposed ADC pipeline demonstrates promising results for categorical labeling tasks, it has a limitation that is important to acknowledge. Currently, the pipeline is specifically designed for categorical labeling. A natural direction for future work is to expand the pipeline's scope to support a broader range of tasks, including object detection and segmentation.

## 6 CONCLUSION

In this paper, we introduced the Automatic Dataset Construction (ADC) pipeline, a novel approach for automating the creation of large-scale datasets with minimal human intervention. By leveraging Large Language Models for detailed class design and automated sample collection, ADC significantly reduces the time, cost, and errors associated with traditional dataset construction methods. The Clothing-ADC dataset, which comprises one million images with rich category hierarchies, demonstrates the effectiveness of ADC in producing high-quality datasets tailored for complex research tasks. Despite its advantages, ADC faces challenges such as label noise and imbalanced data distributions. We addressed these challenges with open-source tools for error detection and robust learning. Our benchmark datasets further facilitate research in these areas, ensuring that ADC remains a valuable tool for advancing machine learning model training.

**Ethical Statement** The Automatic Dataset Construction (ADC) pipeline emphasizes ethical data usage, relying only on publicly available sources that comply with copyright and privacy regulations. Transparency in data handling and clear disclosures are prioritized, encouraging ethical considerations when using ADC, especially in sensitive domains.

**Reproducibility Statement** We ensure reproducibility by providing open-source code for the ADC pipeline when published. In the paper, we give detailed descriptions of data collection (Section 2), label noise mitigation (Section 3), and class balancing (Section 4). All experiments follow standard models, with full documentation of experimental setups and evaluation metrics in the Appendix C. The reproducible code is uploaded as well.

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

# A APPENDIX

## APPENDIX

The appendix is organized as follows:

- Appendix A includes additional detailed algorithms in the Automatic-Dataset-Construction pipeline.
- Appendix B contains dataset statistics and more exploratory data analysis of Clothing ADC.
- Appendix C includes experiment details of our benchmark on label noise detection, label noise learning, and class-imbalanced learning.

### BROADER IMPACTS

Our paper introduces significant advancements in dataset construction methodologies, particularly through the development of the Automatic Dataset Construction (ADC) pipeline:

- **Reduction in Human Workload:** ADC automates the process of dataset creation, significantly reducing the need for manual annotation and thereby decreasing both the time and costs associated with data curation.
- **Enhanced Data Quality for Research Communities:** ADC provides high-quality, tailored datasets with minimal human intervention. This provides researchers with datasets in the fields of label noise detection, label noise learning, and class-imbalanced learning, for exploration as well as fair comparisons.
- **Support for Customized LLM Training:** The ability to rapidly generate and refine datasets tailored for specific tasks enhances the training of customized Large Language Models (LLMs), increasing their effectiveness and applicability in specialized applications.

Furthermore, the complementary software developed alongside ADC enhances these impacts:

- **Data Curation and Quality Control:** The software aids in curating and cleaning the collected data, ensuring that the datasets are of high quality that could compromise model training.
- **Robust Learning Capabilities:** It incorporates methods for robust learning with collected data, addressing challenges such as label noise and class imbalances. This enhances the reliability and accuracy of models trained on ADC-constructed datasets.

Together, ADC and its accompanying software significantly advance the capabilities of machine learning researchers and developers by providing efficient tools for high-quality customized data collection, and robust training.

### LIMITATIONS

While ensuring the legal and ethical use of datasets, including compliance with copyright laws and privacy concerns, is critical, our initial focus is on legally regulated and license-friendly data sources available through platforms like Google or Bing. Addressing these ethical considerations is beyond the current scope but remains an essential aspect of dataset usage.

Besides, similar to Traditional-Dataset-Construction (TDC), Automatic-Dataset-Construction (ADC) is also unable to guarantee fully accurate annotations.

# A  DETAILED ALGORITHMS IN THE GENERATION OF AUTOMATIC-DATASET-CONSTRUCTION

## A.1  THE ALGORITHM OF IMAGE DATA COLLECTION IN ADC

---
**Algorithm 2** Image Data Collection in ADC

---
1: **procedure** IMAGEDATACOLLECTION
2:     **Part A:** Get attributes from dataset design
3:     **attributes** ← Step 1 Dataset Design
4:     **categories** ← ["sweater", "shirt", "pants", ...]         ▷ List of categories
5:     **target_category** ← "sweater"         ▷ Target category (e.g. "sweater")
6:     **attributes** ← $attributes$[**target_category**]     ▷ Get attributes for target category
7:     $colors, patterns, materials$ ← **attributes**["color"],
8:                            **attributes**["pattern"],
9:                            **attributes**["material"]
10:     **Part B:** Create search queries
11:     **search_queries** ← $\{\, c + p + m + $ **target_category** $\mid$
12:                       $c \in$ **colors**,
13:                       $p \in$ **patterns**,
14:                       $m \in$ **materials**$\}$     ▷ (e.g. "beige fisherman cotton sweater")
15:     **Part C:** Launch distributed image search
16:     $image\_data$ ← distributed_search($search\_queries$,
17:                          $api =$ Google_Images | Bing_Images,
18:                          $n\_process = 30$)
19: **end procedure**

---

## A.2  THE ALGORITHM OF LEARNING-CENTRIC CURATION METHOD IN ADC

---
**Algorithm 3** Learning-centric curation (early-learning memorization behavior)

---
1: **procedure** EARLYSTOPCE(noisyDataset, percentage=25%)
2:     **Part A:** Train classifier over the dataset and apply early stopping
3:     $\mathcal{D}$ ← Load training data         ▷ (images and labels)
4:     $model$ ← Initialize neural network model         ▷ (e.g. ResNet)
5:     $loss\_fn$ ← Define loss function         ▷ (e.g. cross-entropy)
6:     $optimizer$ ← Choose optimizer         ▷ (e.g. SGD, Adam)
7:     **for** $epoch = 1$ to $E \in \{1, 2\}$ **do**
8:         $model$ ← Trainer($\mathcal{D}, loss\_fn, optimizer$)
9:     **end for**
10:     **Part B:** Record predictions and confidence levels
11:     **for** $batch$ in $\mathcal{D}$ **do**
12:         $images$ ← Get batch of images
13:         $outputs$ ← Forward pass: $model(images)$
14:         $confidence$ ← Get confidence levels: softmax($outputs$)
15:     **end for**
16:     **Part C:** Remove samples with lowest $x\%$ confidence level
17:     $threshold$ ← Calculate threshold: percentile($confidence, 100 - x$)
18:     $\mathcal{D}$ ← Filter out samples with confidence below $threshold$
19:     **Return** $\mathcal{D}$
20: **end procedure**

---

# B  DATASET STATISTICS IN CLOTHING-ADC

## B.1  COLLECTED CLOTHING ADC DATASET

Our collected Clothing-ADC dataset can be found here: Google Drive.

## B.2 ATTRIBUTES CANDIDATES IN CLOTHING-ADC

Our automated dataset creation pipeline is capable of generating numerous designs per attribute, as shown in Table 6. This table provides a detailed list of designs generated by our pipeline, from which we selected a subset to include in our dataset.

| Color | | | Material | | | Pattern | | | | | |
|---|---|---|---|---|---|---|---|---|---|---|---|
| Animal print | Gold | Pastel | Acrylic | Lace | Tulle | Abstract | Camouflage | Fishnet | Leather | Printed | Thongs |
| Beige | Gray | Peach | Alpaca | Leather | Tweed | Abstract Floral | Chalk stripe | Floral | Logo | Quilted | Tie-Dye |
| Black | Green | Pink | Angora | Lightweight | Twill | Animal Print | Check | Floral print | Low rise | Reversible | Tie-dye |
| Blue | Grey | Plum | Bamboo | Linen | Velvet | Animal print | Checkered | Fringe | Mesh | Ribbed | Toile |
| Blush Pink | Heather | Purple | Breathable | Mesh | Viscose | Aran | Chevron | G-strings | Military | Ripples | Tribal |
| Bright Red | Ivory | Red | Cashmere | Microfiber | Water-resistant | Argyle | Color block | Galaxy | Mock turtleneck | Satin | Trench |
| Brown | Khaki | Rich Burgundy | Chambray | Modal | Windproof | Aztec | Colorblock | Garter Stitch | Mosaic | Scales | Tuck stitch |
| Burgundy | Lavender | Royal Blue | Chiffon | Mohair | Wool | Basket check | Cotton | Garter stitch | Moss stitch | Seamless | Tweed |
| Burnt Orange | Light Grey | Rust | Corduroy | Neoprene | bamboo | Basket rib | Cropped | Geometric | Moto | Seed stitch | Twill |
| Champagne | Maroon | Rustic Orange | Cotton | Nylon | cotton | Basket weave | Damask | Gingham | Nailhead | Shadow stripe | Vintage-inspired |
| Charcoal | Metallic | Sage | Crochet | Organza | hemp | Basketweave | Denim | Glen check | Nehru | Sharkskin | Waterproof |
| Charcoal Grey | Mustard | Silver | Denim | PVC | linen | Batik | Diagonal grid | Gradient | Nordic | Sherpa | Windowpane |
| Cream | Mustard Yellow | Soft Pink | Down | Polyester | lycra | Bikini | Diamond | Graphic | Ombre | Silk | |
| Cream White | Navy | Striped | Embroidered | Rayon | modal | Birdseye | Ditsy | Grid | Oversized | Slip Stitch | |
| Dark Plum | Navy Blue | Tan | Flannel | Reflective | nylon | Blazer | Dogtooth | Herringbone | Oxford | Slip stitch | |
| Deep Blue | Neon | Teal | Fleece | Ripstop | polyester | Bomber | Embossed | High waisted | Paisley | Solid | |
| Deep Purple | Nude | Turquoise | Fringe | Satin | rayon | Boxer briefs | Embroidered | Honeycomb | Peacoat | Striped | |
| Earthy Beige | Olive | Vibrant Turquoise | Fur | Silk | silk | Briefs | Emoji | Houndstooth | Pin Dot | Stripes | |
| Floral | Olive Green | Warm Brown | Gore Tex | Softshell | spandex | Brioche | Entrelac | Ikat | Pinstripe | Studded | |
| Forest Green | Orange | White | Gore-Tex | Spandex | tencel | Broken rib | Eyelet | Intarsia | Plaid | Suede | |
| Fuchsia | Pale Yellow | Yellow | Hemp | Suede | viscose | Broken stripe | Fair Isle | Jacquard | Polka Dot | Tartan | |
| | | lilac | Insulated | Synthetic | wool | Cable | Fibonacci | Knit and Purl | Polka dot | Teddy | |
| | | | Jersey | Synthetic Blend | | Cable knit | Fisherman | Lace | Prince of Wales | Textured | |
| | | | Knit | Tencel | | | | | | | |

Table 6: The union of attributes across all clothing types in Clothing-ADC dataset.

## B.3 HUMAN-IN-THE-LOOP CURATION FOR CLOTHINGADC TESTSET

Our automated dataset collection pipeline enabled us to create a large, noisy labeled dataset. We asked annotators to select the best-fitting options from a range of samples, as shown in Figure 5, with each task including at least 4 samples and workers completing 10 tasks per HIT at a cost of $0.15 per task, totaling $150 estimated wage of $2.5-3 per hour, and after further cleaning the label noise, we ended up with 20,000 samples in our test set. To participate, workers had to meet specific requirements, including being Master workers, having a HIT Approval Rate above 85%, and having more than 500 approved HITs, with the distribution of worker behavior shown in Figure 6.

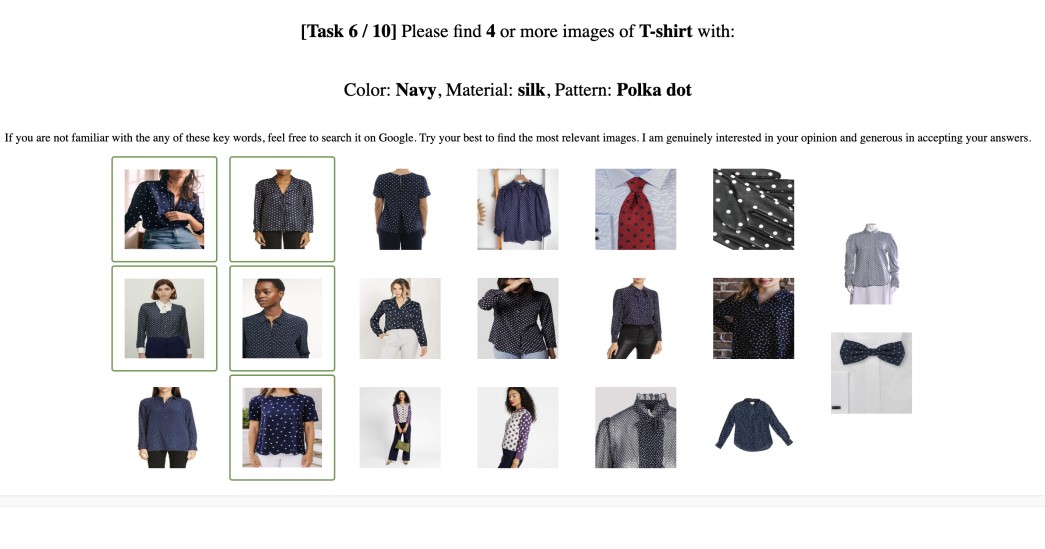

Figure 5: Collection of Clothing-ADC test set: A filtering task to the worker instead of annotation from scratch.

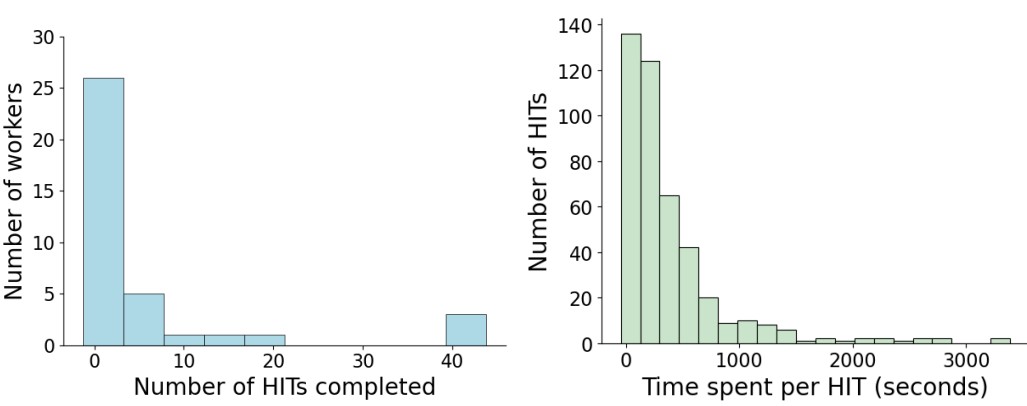

(a) Distribution of the HITs completed per worker      (b) Distribution of work time in seconds per HIT.

Figure 6: The behaviors of workers in the creation of test set.

## B.4 COST ANALYSIS FOR CLOTHINGADC HUMAN-IN-THE-LOOP DATA CURATION

When clean data is required, we recommend combining human involvement with algorithmic approaches to ensure high accuracy. We collected 20,000 samples for both the test set and evaluation set, ensuring a robust and reliable dataset.

We evaluate human effort in Table 7. We used the number of mouse clicks required for each label, excluding overhead costs due to different layout designs across datasets. While other metrics like time spent or monetary cost could be used within the same dataset, they are not easily comparable across datasets with different setups and participants.

| Dataset | Class Count | Noise Rate | Label per Sample | Cost per Label (Click) | Total Cost ($) | Samples Collected |
|---|---|---|---|---|---|---|
| ClothingADC Testset | 12k | Clean | 4 | **0.25** | $150 / 150 | 20k / 20k |
| Cifar-10 N | 10 | ∼18% | 1 | 3 | $450 | 50k |
| Cifar-100 N | 100 | ∼40% | 1 | 1 | $700 | 50k |
| Cifar-10 H | 10 | 5% | 1 | 50 | $3,856.5 | 20k |

Table 7: Human Effort Comparison with Existing Label Noise Datasets.

## B.5 "CLEAN SET" FROM TRADITIONAL METHODS IS NOT ALWAYS CLEAN

The noise rate in the manually annotated dataset iNaturalist is close to 0, suggesting that traditional methods requiring experts are more robust than our proposed ADC pipeline. However, we would like to cite Northcutt et al. (2021b) that even well-curated and widely-adopted "clean" test datasets, which have invested significant effort in ensuring data quality, may still contain errors [2]. This highlights that achieving a 0% noise rate is extremely challenging, even with expert annotation. The table below is the evidence of such observations (from Table 2 in Northcutt et al. (2021b)).

Moreover, a "fully-cleaned" set typically consumes much more time and money. When the budget is limited, the annotation accuracy is much lower. For example, the collection of CIFAR-10N Wei et al. (2022b), where each training image of CIFAR-10 (a relatively easy 10-class classification) is assigned to 3 independent annotators. To collect 3 annotations for each of the 50K images, it takes >2 days and >1000 dollars on Amazon Mturk. However, the overall annotation error is approximately 18%. As for CIFAR-100N Wei et al. (2022b), this is a much more challenging task where each annotator is requested to find out the most relevant label for each image among 100 classes (50K images in all). It takes >2 days and > 800 dollars on Amazon Mturk. However, the overall annotation error is approximately 40%.

---

[2]https://labelerrors.com/

| Dataset (Test Set) | Size | % Error |
|---|---|---|
| MNIST | 10000 | 0.15 |
| CIFAR-10 | 10000 | 0.54 |
| CIFAR-100 | 10000 | 5.85 |
| Caltech-256 | 29780 | 1.84 |
| ImageNet | 50000 | 5.83 |
| QuickDraw | 50426266 | 10.12 |
| 20News | 7532 | 1.09 |
| IMDB | 25000 | 2.90 |
| Amazon Reviews | 9996437 | 3.90 |
| AudioSet | 20371 | 1.35 |

Table 8: Error comparison across datasets (from Table 2 in Northcutt et al. (2021b))

## C  EXPERIMENT DETAILS

### C.1  DISTRIBUTION OF HUMAN VOTES FOR LABEL NOISE EVALUATION

On the annotation page, we presented the image and its original label to the worker and asked if they believed the label was correct (Figure 7). They input their evaluation by clicking one of three buttons. Note that we encouraged workers to categorize acceptable samples as "unsure". The resulting distribution is shown in Table 9. Using a simple majority vote aggregation, we found that the noise rate in our dataset is 22.15%. However, if a higher level of certainty is required for clean labels, we can apply a more stringent aggregation method, considering more samples as mislabeled. In the extreme case where any doubts from any of the three annotators can disqualify a sample, our automatically collected dataset still retains 61.25% of its samples.

For the label noise evaluation task, we utilized a subset of 20,000 samples from the Clothing-ADC dataset, collecting three votes from unique workers for each sample. Each Human Intelligence Task (HIT) included 20 samples and cost $0.05. To participate, workers had to meet the following requirements: (1) be Master workers, (2) have a HIT Approval Rate above 85%, and (3) have more than 500 approved HITs. The total cost for this task was $150, estimated wage of $2.5-3 per hour.

We show the distribution of worker behavior during the noise evaluation task in Figure 8. Figure 8(a) shows the distribution of the amount of HIT completed per worker while neglecting ids with 1-2 submissions. There is a total of 49 unique workers. Figure 8(b) shows the distribution of time spent per HIT.

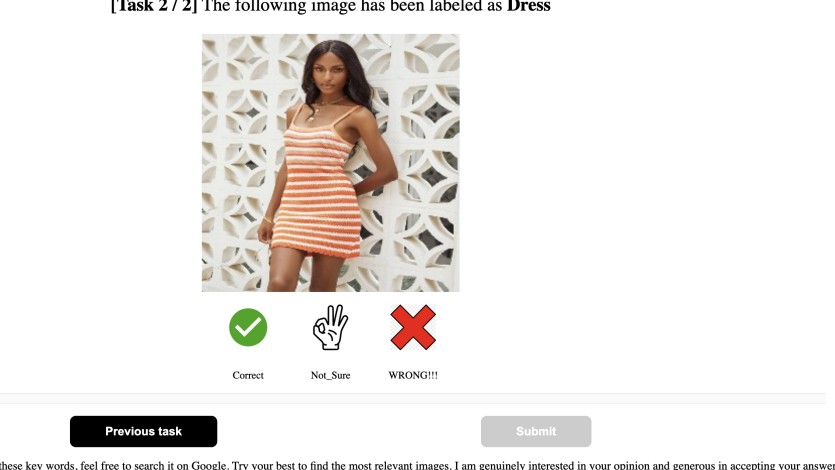

Figure 7: Label noise evaluation worker page

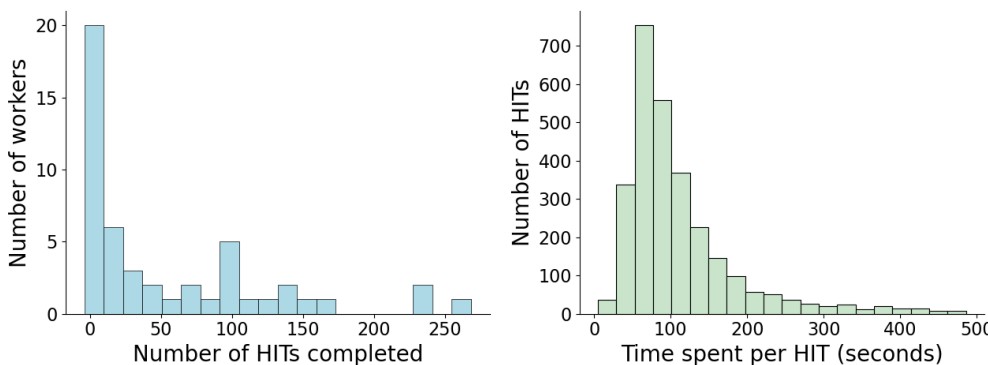

(a) Distribution of the HITs completed per worker    (b) Distribution of work time in seconds per HIT.

Figure 8: The behaviors of workers in the collection of label noise evaluation.

Table 9: **Distribution of Human Votes for Label Noise Evaluation**: We employed human annotators to evaluate a subset of 20,000 samples from our collected dataset, with each sample receiving three votes from distinct annotators.

| Human Votes | Percentage |
|---|---|
| Yes, Yes, Yes | 61.25% |
| Yes, Yes, Unsure | 6.10% |
| Yes, Yes, No | 10.50% |
| Else | 22.15% |

## C.2 NOISY LEARNING AND CLASS IMBALANCE LEARNING BENCHMARK IMPLEMENTATION DETAILS

Our code refers to zip file in supplementary material.

```python
train_set = Clothing1mPP(root, image_size, split="train")
tiny_set_ids = train_set.get_tiny_ids(seed=0)
tiny_train_set = Subset(train_set, tiny_set_ids) # Get the tiny version
    of the dataset
val_set = Clothing1mPP(
    root, image_size, split="val", pre_load=train_set.data_package
)
test_set = Clothing1mPP(
    root, image_size, split="test", pre_load=train_set.data_package
)

train_loader = DataLoader(
    train_set, batch_size=batch_size, shuffle=True, num_workers=
    num_workers
)
tiny_train_loader = DataLoader(
    tiny_train_set, batch_size=batch_size, shuffle=True, num_workers=
    num_workers
)
val_loader = DataLoader(
    val_set, batch_size=batch_size, shuffle=False, num_workers=
    num_workers
)
test_loader = DataLoader(
    test_set, batch_size=batch_size, shuffle=False, num_workers=
    num_workers
)
```

Listing 1: How to load data. Line 1 loads the full set of our dataset. Line 2 and Line 3 load the tiny version of our dataset. Line 4 creates the validation set. Line 5 creates the testing set. Line 11 to Line 20 create the data loader.

```
python examples/main.py --config configs/Clothing1MPP/default.yaml # Run
    Cross Entropy
python examples/main_peer.py --config configs/Clothing1MPP/default.yaml #
     Run Peer Loss
python examples/main_jocor.py --config configs/Clothing1MPP/default_jocor
    .yaml # Run Jocor
python examples/main_coteaching.py --config configs/Clothing1MPP/
    default_coteaching.yaml # Run Co-teaching
python examples/main_drops.py --config configs/Clothing1MPP/default_drops
    .yaml # Run drops
```

Listing 2: The example of the command we use to run the algorithm in one line

```yaml
inherit_from: configs/default.yaml
data: &data_default
  root: '/root/cloth1m_data_v3'
  image_size: 256
  dataset_name: "clothing1mpp"
  imbalance_factor: 1 # 1 means no imbalance
  tiny: False

train: &train
  num_workers: 8
  loss_type: 'ce'
  loop_type: 'default' # 'default','peer','drops'
  epochs: 20
  global_iteration: 999999999
```

```
15    batch_size: 64
16    # scheduler_T_max: 40
17    scheduler_type: 'step'
18    scheduler_gamma: 0.8
19    scheduler_step_size: 2
20    print_every: 100
21    learning_rate: 0.01
22
23  general:
24    save_root: './results/'
25    whip_existing_files: True # Whip exisitng files
26    logger:
27      project_name: 'Clothing1MPP'
28      frequency: 200
29
30  model: &model_default
31    name: "resnet50"
32    pretrained_model: 'IMAGENET1K_V1'
33    cifar: False
34
35  test: &test_defaults
36    <<: *train
```

Listing 3: The example of YAML config file

### C.3 LABEL NOISE DETECTION BENCHMARK

We run four baselines for label noise detection, including CORES Cheng et al. (2020), confident learning Northcutt et al. (2021a), deep $k$-NN Papernot & McDaniel (2018) and Simi-Feat Zhu et al. (2022). All the experiment is run for one time following Cheng et al. (2020); Zhu et al. (2022).

The experiment platform we run is a 128-core AMD EPYC 7742 Processor CPU and the memory is 128GB. The GPU we use is a single NVIDIA A100 (80GB) GPU. For the dataset, we used human annotators to evaluate whether the sample has clean or noisy label as mentioned in Appendix C.1. We aggressively eliminates human uncertainty factors and only consider the case with unanimous agreement as a clean sample, and everything else as noisy samples. The backbone model we use is ResNet-50 He et al. (2016). For all the baselines, the parameters we use are the same as the original paper except the data loader. We skip the label corruption and use the default value from the original repository. For CORES, the cores loss whose value is smaller than 0 is regarded as the noisy sample. For confidence learning, we use the repository[3] from the clean lab and the default hyper-parameter. For deep $k$-NN, the $k$ we set is 100. For SimiFeat, we set $k$ as 10 and the feature extractor is CLIP.

### C.4 LABEL NOISE LEARNING BENCHMARK

The platform we use is the same as label noise detection. The backbone model we use is ResNet-50 He et al. (2016). For the full dataset, we run the experiment for 1 time. For the tiny dataset, we run the experiments for 3 times. The tiny dataset is sampled from the full set whose size is 50. The base learning rate we use is 0.01. The base number of epochs is 20. The hyper-parameters for each baseline method are as follows. For **backward and forward correction**, we train the model using cross-entropy (CE) loss for the first 10 epochs. We estimate the transition matrix every epoch from the 10th to the 20th epoch. For the **positive and negative label smoothing**, the smoothed labels are used at the 10th epoch. The smooth rates of the positive and negative are 0.6 and -0.2. Similarly, for **peer loss**, we train the model using CE loss for the first 10 epochs. Then, we apply peer loss for the rest 10 epochs and the learning rate we use for these 10 epochs is 1e-6. The hyper-parameters for $f$-**div** is the same as those of peer loss. For **divide-mix**, we use the default hyper-parameters in the original paper. For **Jocor**, the hyper-parameters we use is as follows. The learning rate is 0.0001. $\lambda$ is 0.3. The epoch when the decay starts is 5. The hyper-parameters of **co-teaching** is similar to Jocor. For logitclip, $\tau$ is 1.5. For **taylorCE**, the hyper-parameter is the same as the original paper.

---

[3]https://github.com/cleanlab/cleanlab

## C.5 CLASS-IMBALANCED LEARNING BENCHMARK

The platform we use is the same as label noise detection. The backbone model we use is ResNet-50 He et al. (2016). For different imbalance ratio ($\rho = 10, 50, 100$). The class distribution is shown in Table 10. For all the methods, the base learning rate is 0.0001 and the batch size is 448. The dataset we use is not full dataset because we want to disentangle the noisy label and class imbalance learning. We use Docta and a pre-trained model trained with cross-entropy to filter the data whose prediction confidence is low. Due to the memorization effect, we fine-tune the model for 2 epochs to filter the data. We remove 45.15% data in total where Docta removes 26.36% while CE removes 25.00% with a overlap of 6.20%. Thus, the datset we use for class-imbalance learning is 54.85% of the full dataset.

| imbalance ratio ($\rho$) | Class Distribution | Total Number |
|---|---|---|
| 10 | [39297, 31875, 25854, 20971, 17010, 13797, 11191, 9078, 7363, 5972, 4844, 3929] | 191181 |
| 20 | [39297, 27536, 19295, 13520, 9474, 6638, 4652, 3259, 2284, 1600, 1121, 785] | 129461 |
| 100 | [39297, 25854, 17010, 11191, 7363, 4844, 3187, 2097, 1379, 907, 597, 392] | 114118 |

Table 10: The class distribution for different imbalance ratio

## D DEMO APPLICATION OF ADC IN OTHER FIELDS

Our Automated Dataset Construction (ADC) pipeline is best suited for image classification tasks where the relevant knowledge can be easily searched and retrieved from the internet. Example applications include, but are not limited to:

- Food classification
- Hairstyle classification
- Vehicle classification
- Home decor classification
- Plant classification
- Sport equipment classification
- Jewelry classification

**Food Classification** To illustrate the effectiveness of our ADC pipeline, let's consider a more detailed example of food classification. We used the prompt "Food Classification: Create a dataset with various types of cuisine, and sub-classes for specific dishes, ingredients, or cooking methods. Help me to find 10 different attributes to describe food." LLM generated a range of subcategories to describe different types of food, including, but are not limited to:

- Cuisine type (Italian, Chinese, Indian, etc.)
- Dish Type (Appetizer, main course, dessert, etc.)
- Protein source (Beef, Chicken, Tofu, etc.)
- Cooking method (Grilled, Baked, Fried, etc.)
- Spice level (Mild, Medium, Spicy, etc)
- Allergen warning (Gluten-free, Nut-free, Dairy-free, etc.)
- Texture (Crunchy, Chewy, Smooth, etc)

Please feel free to use the prompt on your favorite LLMs, or modify it slightly for other tasks that interest you more. We tried various LLM versions from OpenAI, Meta, Google, and Claude, and all of them are competent to solve this task, albeit with different preferences for suggesting labels and descriptions.

# E    COPYRIGHT ISSUE

One possible approach to mitigate the potential copyright issues is to rely on the advanced features in search engines provided by the leading industry companies. For example, we can use the "Advanced Image Search => usage rights" function in Google Image Search, which allows users to filter search results by usage rights.

However, We must clarify that our pipeline is provided "as-is" and that users are responsible for using the collected data at their own risk. We cannot guarantee that the data is free from copyright issues, and users must take their own steps to ensure compliance with applicable laws and regulations. This approach is similar to that taken by the LAION-5B dataset Schuhmann et al. (2022), which states that "The images are under their copyright."

