# OpenReview forum: "Automatic Dataset Construction (ADC): Sample Collection, Data Curation, and Beyond"
_ICLR.cc/2025/Conference — Submitted to ICLR 2025_

### Official Review · Reviewer_6pyT · 2024-10-29

**Soundness:** 3
**Presentation:** 3
**Contribution:** 3
**Rating:** 6
**Confidence:** 3

**Summary:**

In this paper, the authors proposed propose Automatic Dataset Construction (ADC), a methodology that automates dataset creation with negligible cost and high efficiency. ADCleverages LLMs for the detailed class design and code generation to collect relevant samples via search engines. The authors design three benchmark datasets focused on label noise detection, label noise learning, and class-imbalanced learning.

**Strengths:**

1. The proposed Automatic Dataset Construction (ADC) leverages LLMs for the detailed class design and code generation to collect relevant samples via search engines, reducing the need for manual annotation and speeding up the data generation process.

2.  The authors explore several challenges observed in real-world dataset construction, including detecting label errors, learning with noisy labels, and class-imbalanced learning.

**Weaknesses:**

Could the authors please address the following questions:

1. In Step 2 of Fig. 1, the explanation for "image search engine" states, "Labeling, ADC reduces human workload by flipping the data collection process, using targets to search for samples." What is the conceptual meaning of this part?

2. In Step 3 of Fig. 1, what is the insight behind using "filter" instead of "relabeling"?

**Questions:**

Please the Weaknesses above.

---

> ### Author Response · Authors · 2024-11-18
>
> We thank the reviewer for thoughtful comments.
>
> **Weakness 1: In Step 2 of Fig. 1, the explanation for "image search engine" states, "Labeling, ADC reduces human workload by flipping the data collection process, using targets to search for samples." What is the conceptual meaning of this part?**
>
> Traditional dataset construction(TDC) first locates a set of unlabeled images, and second asks human annotators to manually provide a classification label for each image. In our example dataset this would mean finding 1,000,000 images of clothing, and then for each image having a human indicate which class attribute labels apply (sweater, beige, wool, …).
>
> Our method (ADC) flips the order, first developing a set of class attribute labels, and second using an image search engine to search for samples which specifically belong to a set of labels. Flipping the order in this way reduces the requirement to manually label each of the 1,000,000 images collected.
>
> **Weakness 2: Clarification on the ADC label cleaning in step 3**
>
> **Why filter:** The efficiency of the ADC cleaning comes from the initial data collection. From step 2, ADC resulted in a much larger dataset without human labeling efforts compared to TDC. A large prelabeled-set allows ADC to aggressively remove noisy labeled samples while maintaining the resulting dataset scale. The prelabeled-set is also human effort free, thus filtering samples in ADC won’t waste any human labeling efforts, which is required in the traditional setup.
>
> **Auto filter:** For applications where some label noise can be tolerated, we employ label noise detection methods like [1] to significantly reduce errors. These methods can effectively identify and correct noisy labels, resulting in a more accurate dataset [2]. We found a **79.0%** agreement between automated and human-in-the-loop curation methods. Additionally, we observed a reduction in label noise from **22.2%** to **10.7%** after applying the ADC auto-curation, demonstrating its effectiveness in improving dataset quality.
>
> **Human-in-the-loop filter:** For domains requiring clean data, we advocate for human involvement in addition to algorithmic approaches to ensure perfect annotations. Unlike traditional pipelines where humans are asked to label samples from scratch, our ADC pipeline provides a large amount of noisy labeled samples for humans to review and select the accurate ones. This approach is mentally easier (and thus faster) for the human annotator and results in a very clean dataset, since the selected samples have guaranteed human and machine label agreements. Appendix Table 7 shows a detailed cost analysis.
>
> [1] Zhu, Zhaowei, Yiwen Song, and Yang Liu. "Clusterability as an alternative to anchor points when learning with noisy labels." International Conference on Machine Learning. PMLR, 2021.
>
> [2] Zhu, Zhaowei, Jialu Wang, Hao Cheng, and Yang Liu. "Unmasking and Improving Data Credibility: A Study with Datasets for Training Harmless Language Models." In The Twelfth International Conference on Learning Representations, 2024.

---

### Official Review · Reviewer_kh2s · 2024-10-31

**Soundness:** 3
**Presentation:** 3
**Contribution:** 2
**Rating:** 5
**Confidence:** 4

**Summary:**

This paper introduces Automatic Dataset Construction (ADC) a new methodology to create large-scale datasets with reduced costs and improved efficiency and accuracy. This approach leverages large language models (LLMs) for class design and sample collection, then uses various existing label error detection and robust learning techniques to enhance the quality of the dataset. The authors introduced a new dataset curated using the previously described methods, Clothing-ADC. They also developed versions of the dataset to facilitate benchmarks in label noise detection, label noise learning, and class-imbalanced learning, which they also presented comprehensive results and evaluations for.

**Strengths:**

1. New methodology proposed that will enable efficient dataset construction across many domains, making it adaptable for broad future applications
2. Creation of Clothing-ADC dataset which supports future work on label noise and class imbalance
3. The paper acknowledges challenges that ADC encounters when it comes to ensuring data quality, and the authors present and benchmark a comprehensive set of tools that could help to address these limitations, which helps to make the approach more viable

**Weaknesses:**

1. Lack of novelty in data quality methods. The paper and pipeline relies heavily on existing methods to handle label noise and class imbalance, and does not introduce any new methodologies. It mainly offers a procedural contribution rather than advancing new techniques for improving label noise or class imbalance handling
2. The paper presents many algorithms to identify label noise but is lacking detailed analysis of the impact of each algorithm on the resulting dataset and downstream model performance
3. Related to the point above, the paper does not provide quality metrics to evaluate the quality of the dataset after before/after using the ADC pipeline to clean the data, this makes it hard to assess the effectiveness of the automated curation process and ADC’s ability to produce high quality data

**Questions:**

1. How well will the human-in-the-loop step would scale as the dataset size gets larger, or if there is high levels of noise? Have the authors considered any ranking mechanism to prioritize samples for human review, or as another automatic filtering technique?
2. The current pipeline only checks for low-quality data in the form of noisy labels and imbalanced classes, have the authors considered how ADC could also address other data issues such as outliers, duplicates, and other low quality data (for images, that could include blurry images, odd aspect ratios etc)?

**Details Of Ethics Concerns:**

The paper states that it has ethical data usage through only relying on publicly available sources that comply with copyright and privacy regulations. However I am uncertain if this is adequate to address all potential concerns of web sourcing data, hence am requesting for additional review.

---

> ### Author Response · Authors · 2024-11-18
> **Response 1/2**
>
> We thank the reviewer for thoughtful comments, both pros and cons, and hope to convince you the paper's strengths are more important than the weaknesses. This paper has three parts: a new workflow, a new dataset, and a benchmark. This is somewhat confusing since any one of these could be the subject of a paper. Thus let us explain our reasoning for combining them briefly before answering your specific questions.
>
> Collecting data sets is really hard work. That's one of the reasons why there is a dataset track at conferences now, to acknowledge that a good dataset is worth publishing based on its own value. The traditional method of data set collection - locating a lot of data first and then labeling each sample afterwards - is time consuming, expensive, and produces imbalanced datasets. This paper attempts to address these concerns. It contributes a workflow that dramatically lowers the time, expense and resulting class imbalance. Naturally readers will wonder if the method works. In order to show that this workflow is effective, we introduce a new dataset built using the method. This new dataset has 1M samples, many more classes than most other datasets, and much better class balance than existing datasets. We think this dataset itself will be of interest to some readers. However we expect other readers to object that cheap automatic labeling produces a poor quality dataset. Thus we provide some analysis of noise and a benchmark to show that the dataset does indeed have value to researchers.
>
> **Weakness 1: Lack of novelty in data quality methods**
>
> We completely agree that we offer a procedural contribution rather than introducing new methodologies. We put together existing methods in a new way to dramatically improve the process of dataset creation. We believe this is entirely appropriate for a dataset track. If we were introducing new techniques for improving label noise we would submit that to a research track. There is of course the question of whether our process for creating a dataset is truly novel. We believe it is, since we have created a dataset with better ‘specs’ than other similar datasets. However we don’t know every paper and welcome citations to related papers or datasets which suggest a substantially similar pipeline.
>
> **Weakness 2: Lack of detailed analysis on label noise methods**
>
> For noisy learning dataset, with the increment of dataset size, there is an obvious improvement for the performance. All the noisy learning methods show their effectiveness over the baseline method (Cross-Entropy). TaylorCE achieves the highest performance on both settings, showing the effectiveness of the differential elements. Divide-Mix is also a competitive method because of its dual model architecture. The forward and backward correction is simple but effective.
>
> For class-imbalance unlearning,  Logits-Adjust and Drops are most robust methods for all $\delta$ and $\rho$ because Logit-Adjust directly target the minority class by balancing the decision boundary while Drops incorporate the DRO framework by including a term for worst-case class performance.
>
> **Weakness 3: Lack of metric to evaluate the quality of ADC cleaning**
>
> We agree on the necessity to evaluate the quality of ADC's automated curation process. To address this, we provide quantitative metrics in Section 3.1 through a benchmark comparison of label noise detection methods. To validate ADC's effectiveness, we conducted a human-in-the-loop evaluation, comparing automated curation results against human curation. This analysis revealed a **79.0%** agreement rate between automated and human curation decisions. Additionally, we observed a reduction in label noise from **22.2%** to **10.7%** after applying the ADC auto-curation, demonstrating its effectiveness in improving dataset quality.

---

> ### Author Response · Authors · 2024-11-18
> **Response 2/2**
>
> **Question 1: How well will the human-in-the-loop step would scale as the dataset size gets larger, or if there are high levels of noise? Have the authors considered any ranking mechanism to prioritize samples for human review, or as another automatic filtering technique?**
>
> We used human-in-the-loop for creating a test set with clean human vetted labels. Since we wanted this set to be randomly sampled (with matching statistical distribution to the set as a whole), we did not try to prioritize samples for human review. We think the automatic cleaning methods discussed are sufficient to reduce noise to a level comparable to other datasets, so we were not imaging human review of an entire dataset. However if one wanted to create a dataset with cleaner labels then we agree that the suggestion to prioritize samples is a good one.
>
> **Question 2: The current pipeline only checks for low-quality data in the form of noisy labels and imbalanced classes, have the authors considered how ADC could also address other data issues such as outliers, duplicates, and other low quality data (for images, that could include blurry images, odd aspect ratios etc)?**
>
> This is a great question which we have discussed, but don’t have any conclusions. It's possible that our pipeline reduces duplicates and blurry images because data is selected by a search engine before we collect it, and search engines work hard to provide only high quality results. On the other hand, it's possible we are accidentally collecting a set of samples which doesn’t contain sufficient blurry images for diverse training samples. Understanding the distribution of our collected data versus a pure random scrape of the web is something we hope to look into in the future.
>
> **Reviewer: Ethics: The paper states that it has ethical data usage through only relying on publicly available sources that comply with copyright and privacy regulations. However I am uncertain if this is adequate to address all potential concerns of web sourcing data, hence am requesting for additional review.**
>
> The ethics of dataset collection from public data are controversial and several high profile legal cases are in progress. Our community is likely to struggle with these issues for the foreseeable future. The authors believe our methods are within the scope of common academic and commercial practice in the ML community, but we welcome a discussion with the ICLR Ethics Review Committee if there are concerns specific to our work.

---

> ### Comment · Reviewer_kh2s · 2024-11-27
>
> Thanks for the detailed response and answer to the questions. I acknowledge that good datasets are crucial in enabling future research, however it still lacks of deeper novelty and detailed impact (there is still little analysis of the downstream benefits). I am updating my rating to *5: marginally below the acceptance threshold* as I agree that the dataset itself could be a good contribution.

---

> ### Author Response · Authors · 2024-11-30
>
> Dear reviewer kh2s,
>
> Thank you for reviewing our work thoroughly. We'd like to provide some clarifications of the novelty and impact of our work.
>
>
> # Novelty
>
> ## ADC pipeline
> ADC is the first work to propose automatic dataset construction that works with LLM, which drastically reduces human effort compared to traditional methods(TDC).
>
>
> ## Software efforts
>
> **[Data Curation]:** Raw data collected through ADC or human manual annotations contains label noise. ADC employs label noise detection methods for data curation, which reduced label noise from **≈22.2%** to **≈10.7%** in ClothingADC.
>
> **[Robust Learning Methods]:** Despite efforts from the label noise community, both ADC and TDC result in imperfect training sets. Even well-studied datasets like CIFAR-10, ImageNet, and MNIST fail to provide completely clean sets (https://labelerrors.com). Thus, robust label noise learning and class imbalance learning algorithms are essential for ADC and TDC datasets.
> We demonstrate software solutions for addressing these two issues in dataset construction, and ADC is compatible with most existing approaches.
>
>
> ## Value of dataset and benchmarking
> Collecting data sets and benchmark evaluation is really hard work. That’s why there is a datasets and benchmarks track at ICLR now, to acknowledge that a good dataset with benchmarking is worth publishing based on its own value.
>
> **[Dataset]:** ClothingADC serves as an application example demonstrating ADC's capability to collect large-scale image datasets at low cost within a short time frame.
>
> **[Benchmark]:** Our benchmarks validate the ADC software solutions. The label noise detection benchmark demonstrates the effectiveness of dataset curation from raw collection. The label noise and class imbalance learning benchmarks show the effectiveness and robustness of these methods when working with imperfect training data.
>
>
>
> # Impact: Compare to Clothing1M
> To our knowledge, we are the first to propose an automatic data collection pipeline that works with LLM. While direct comparisons of dataset collection methods are challenging due to differences in application domains and human annotator setups, we can draw meaningful comparisons with Clothing1M [1], a widely cited dataset (**1400+** citations) that shares similar characteristics with ClothingADC.
>
> Both Clothing1M and ClothingADC contain 1 million web-based clothing images with label noise. However, ClothingADC offers several advantages:
> - **Data Construction Cost:** Clothing1M's training set requires human manual labeling, whereas ClothingADC's training set eliminates the need for human annotation.
> - **Label Noise Quality:** Clothing1M has an overall noise rate of **38.46%**, while ClothingADC achieves significantly lower rates: 22.2% in raw data and 10.7% after our curation process.
> - **Superior Training Performance:**
>     - As demonstrated in Table 4 of our work, label noise learning methods trained on ClothingADC achieve 77.51-81.87% accuracy, showing a substantial improvement of 2.75-7.11% over the CE baseline (74.76%).
>     - In comparison, Wei et al. [2] (Table 6) tested similar label noise learning methods on Clothing1M, achieving 69.13-74.24% accuracy with a smaller improvement of 0.19-5.3% over CE baseline (68.94%​​).
>     - The higher CE baseline accuracy and larger improvements from label noise algorithms suggest that ClothingADC's quality matches or exceeds that of Clothing1M, despite significantly lower dataset collection costs.
>
> **Thanks again, your feedbacks are useful for making our work better.**
>
> Best Regards,
>
> Authors of Submission 7584
>
> **Reference**
>
> [1] Xiao, Tong, et al. "Learning from massive noisy labeled data for image classification." Proceedings of the IEEE conference on computer vision and pattern recognition. 2015.

---

### Official Review · Reviewer_YU7i · 2024-11-01

**Soundness:** 2
**Presentation:** 3
**Contribution:** 2
**Rating:** 5
**Confidence:** 4

**Summary:**

The paper presents the Automatic Dataset Construction (ADC) pipeline, which automates the dataset creation process by leveraging large language models (LLMs) for sample collection, reducing manual annotation efforts, and enhancing efficiency in dataset generation.

**Strengths:**

The paper is well-organized and clearly written, easy to follow.

**Weaknesses:**

1. The paper’s purpose feels somewhat misaligned with practical needs. If the goal is truly Automatic Dataset Construction, the intentional design of three benchmark datasets focusing on label noise detection, label noise learning, and class-imbalance learning seems contradictory. The intent behind this approach isn’t entirely clear to me.

2. The novelty is limited, as the methodology primarily relies on leveraging LLMs to construct the dataset, which feels more like engineering work than a novel research contribution.

3. Lack of clarity regarding how the ADC pipeline addresses potential copyright and ethical concerns associated with using LLMs for data collection.

**Questions:**

1. Could the authors clarify the rationale behind designing specific benchmark datasets for label noise and class imbalance in the context of an automatic dataset construction pipeline? How do these benchmarks align with the overall goal of automation?

2. Given that the ADC pipeline relies on LLMs for dataset construction, what measures are in place to ensure the accuracy and relevance of data collected, especially in cases where human annotation is minimized?

**Details Of Ethics Concerns:**

1. In the data search and dataset construction process using LLMs, have the authors considered security implications? Specifically, how does the system ensure that inappropriate or unethical images are filtered out?

2. How does the ADC pipeline address issues of fairness and reliability in the data it collects? Are there specific measures in place to safeguard ethical standards and ensure data integrity?

3. How do the authors address potential copyright concerns? Are there safeguards in place to ensure that collected data complies with copyright laws?

---

> ### Author Response · Authors · 2024-11-18
> **Response 1/2**
>
> We thank the reviewer for thoughtful comments, both pros and cons, and hope to convince you the paper's strengths are more important than the weaknesses.
>
> **Weakness 1: Why does the paper include dataset construction, dataset, and three benchmarks?**
>
> This paper has three parts: a new workflow, a new dataset, and a benchmark. This is somewhat confusing since any one of these could be the subject of a paper. Thus let us explain our reasoning for combining them.
>
> Collecting data sets is really hard work. That's one of the reasons why there is a dataset track at conferences now, to acknowledge that a good dataset is worth publishing based on its own value. The traditional method of data set collection - locating a lot of data first and then labeling each sample afterwards - is time consuming, expensive, and produces imbalanced datasets. This paper attempts to address these concerns. It contributes a workflow that dramatically lowers the time, expense and resulting class imbalance. Naturally readers will wonder if the method works. In order to show that this workflow is effective, we introduce a new dataset built using the method. This new dataset has 1M samples, many more classes than most other datasets, and much better class balance than existing datasets. We think this dataset itself will be of interest to some readers. However we expect other readers to object that cheap automatic labeling produces a poor quality dataset. Thus we provide some analysis of noise and a benchmark to show that the dataset does indeed have value to researchers.
>
> **Weakness 2: The novelty is limited, as the methodology primarily relies on leveraging LLMs to construct the dataset, which feels more like engineering work than a novel research contribution.**
>
> We completely agree that we offer a procedural methods contribution rather than a new technology. One framing for the research question we want to ask is “How can we construct large scale datasets inexpensively which contain thousands of class labels and maintain balance across classes?” Traditional dataset creation can’t achieve this goal, as evidenced by lack of datasets with these properties. We put together existing methods in a new way to dramatically improve the process of dataset creation. We believe this is entirely appropriate for a dataset track. There is of course the question of whether our process for creating a dataset is truly novel. We believe it is, since we have created a dataset with better ‘specs’ than other similar datasets. However we don’t know every paper and welcome citations to related papers or datasets which suggest a substantially similar pipeline.
>
> **Weakness 3: Lack of clarity regarding how the ADC pipeline addresses potential copyright and ethical concerns associated with using LLMs for data collection.**
>
> We want to clarify that LLMs are used only to help define class labels, not to produce actual data. The actual data collection occurs using search engines such as Google or Bing. However the copyright and ethical concerns remain. We believe our method falls within the scope of common academic practice, and have responded to each of the listed concerns in the Ethics section below.
>
> **Question 2: Given that the ADC pipeline relies on LLMs for dataset construction, what measures are in place to ensure the accuracy and relevance of data collected, especially in cases where human annotation is minimized?**
>
> The ADC pipeline uses LLMs primarily as assistive tools for dataset designers to collect existing data from the internet, rather than generating synthetic samples. The LLM's role is focused on providing categorical options, while image collection and labeling are handled through the image search engine.
>
> For dataset quality assurance, we implemented automated curation methods to refine the collected dataset. The effectiveness of our label noise detection approaches is demonstrated in Section 3.1, where we found a **79.0%** agreement between automated and human-in-the-loop curation methods. Additionally, we observed a reduction in label noise from **22.2%** to **10.7%** after applying the ADC auto-curation, demonstrating its effectiveness in improving dataset quality.
>
> When noise levels below 10% are required, human labels of some kind are needed. This is true for all dataset collection methods. We introduce a method to gather these human judgements with lower cost than asking the human to label each image directly. We used our method to collect a 20k-sample ‘test subset’, with details in Appendix C1.

---

> ### Author Response · Authors · 2024-11-18
> **Response 2/2**
>
> **Ethics Concerns**
>
> **Ethics 1:  How does the system ensure that inappropriate or unethical images are filtered out?**
>
> All images collected have been returned by a ‘search engine’. All major search companies include some sort of ‘safe search’ option when returning images, and we rely on their implementation.
>
> **Ethics 2: How does the ADC pipeline address issues of fairness and reliability in the data it collects?**
>
> All methods which scrape data from the web are subject to the biases that exist in current web data. Traditional dataset construction generally produces a dataset with statistics that match the existing web, resulting in large class imbalance and under-representation of some ideas and groups. ADC is specifically designed to partially correct this bias by building a dataset with many class labels and enforcing class balance across those labels.
>
> **Ethics 3: How do the authors address potential copyright concerns?**
>
> The ethics of dataset collection from public data are controversial and several high profile legal cases are in progress. Our community is likely to struggle with these issues for the foreseeable future. The authors believe our methods are within the scope of common academic and commercial practice in the ML community.
>
> We welcome a discussion with the ICLR Ethics Review Committee if there are concerns specific to our work around any of these issues.

---

### Official Review · Reviewer_8uBb · 2024-11-03

**Soundness:** 3
**Presentation:** 3
**Contribution:** 3
**Rating:** 8
**Confidence:** 3

**Summary:**

The paper tackles the problem of automatic dataset creation. It proposes the ADC pipeline for the same which requires minimal human overhead. The proposal is to first decide the attributes (using a LLM) and then crawl data using attribute as filters (using Google and BING APIs). This is in contrast to existing approaches that label instances given a list of classes. Finally the instances are cleaned to filter noisy labeled samples.

The paper proposes Clothing-ADC dataset with 1M+ samples. Moreover, the paper also discusses the challenges associated with dataset construction including label errors, noisy labels and class imbalance. It then presents a solution (based on existing approaches for noise detection) to tackle these challenges to clean or improve reliability of the constructed dataset.

**Strengths:**

1. The paper is well written and easy to follow.
2. The paper tackles an important problem of dataset creation - the proposed approach (ADC) can do it in cost effective manner. Table 7 provides a comparison against existing label noise datasets including Cifar 10 N / H and Cifar to demonstrate the effectiveness of ADC.
3. The paper presents the Clothing-ADC dataset with 1M+ samples with 12K classes.
4. The paper also presents a subset of Clothing-ADC CLT which is suitable for class imbalance learning. Various baselines (Drops, Bal-softmax, Logit-Adjust) are also reported for this dataset. A subset of 20K samples is also proposed for label noise detection.
5. It is important to discuss the biases (tail or infrequent classes) and noise (wrongly labeled samples) introduced due to the web data - the paper does so clearly.
6. The code and hyper-parameter details are clearly specified.

**Weaknesses:**

1. The ADC methodology is general; however, the paper applies it only to image data (Clothing-ADC dataset). A brief discussion can be included for other domains (such as text), if it applies.
2. The approach is specifically designed for cases when data is fetched as part of process. It may not apply to the cases where data is available in some form such as unlabelled corpus.

**Questions:**

1. Please discuss the applicability of the proposed methodology for other domains such as text. How will it translate - some components (cleaning) are straightforward, others not so much (web scraping or labeling)? Please clarify if it doesn't apply.
2. It will be helpful to have a more elaborate discussion on synthetic datasets. The paper already includes benefits of ADC over TDC. ADC should be clearly contrasted against synthetic data. Can some of the issues introduced by hallucination be recitified in cleaning stage?
3. Please include statistics for class distribution - this will provide a better idea of imbalance as well.
4. The following sentence is a bit unclear "For applications where some label noise can be tolerated, existing data curation software capable of identifying and filtering out irrelevant images, such as Docta, CleanLab , and Snorkel 1, etc."

---

> ### Author Response · Authors · 2024-11-18
> **Response 1/2**
>
> We are happy you believe our work tackles an important problem of dataset creation. Thanks for the constructive feedback, both pro and con. We respond to the listed weaknesses and questions below.
>
> **[W1 and Q1] Additional Application of ADC**
>
> We completely agree that a wider range of applications would make ADC more useful. The paper presented an image classification dataset, and the authors believe ADC can be directly applied to various domains of image classification like clothing, food, sports equipment, etc. (More samples are provided in Appendix D)
>
> In order to expand to new modalities such as text or audio or video, our existing method would need search engines to return a data ‘sample’ that matches the class attribute labels. Current search engines do this, but questions remain on the granularity of the sample. If the goal is a dataset of text samples in the form of complete web pages, then we think the existing method can be applied directly. However if the goal is a dataset of samples at the granularity of a sentence or paragraph, then some additional processing would be needed to turn web page centric search results into data samples to include in a dataset. Some search engines now return ‘highlights’ showing the specific sentence within a webpage that matches the search terms, and this could be used to extract sentence level results. However our answer is speculation only, we focused exclusively on image datasets in our work, and have not tested gathering datasets of other modalities.
>
> **[W2] The approach is specifically designed for cases when data is fetched as part of a process. It may not apply to the cases where data is available in some form such as unlabelled corpus.**
>
> ADC works on well defined concepts with lots of publicly available data, like clothing, food. ADC helps users to create customized datasets in these domains.
>
> Working with a specifically provided unlabeled corpus and restricting the dataset to only this corpus is outside the scope of what we considered, and we agree this is a weakness. However, as a thought exercise for possible future research - We imagine using a pre-trained VLM as a zero-shot classifier to select samples from the corpus which match specific class label attributes. It would also be possible to fine-tuning the VLM with samples and machine labels collected by ADC from outside the corpus, before applying it to select samples from within the corpus. After that you can follow the ADC step 3 for label cleaning, if higher quality is needed in your application.
>
> **[Q2] It will be helpful to have a more elaborate discussion on synthetic datasets. The paper already includes benefits of ADC over TDC. ADC should be clearly contrasted against synthetic data. Can some of the issues introduced by hallucination be rectified in the cleaning stage?**
>
> We completely agree that hallucination is an issue in synthetically generated datasets. This is why we specifically used image search engines to collect real samples from the internet, instead of using image generators like stable diffusion to synthesize new samples. We haven’t done any experiments on synthetic data, but as a thought exercise - Our human based cleaning could clearly be applied to synthetic data. For automated cleaning, one possible direction is to use real samples collected by ADC as anchor points to cross validate the reliability of the synthetic data.

---

> ### Author Response · Authors · 2024-11-18
> **Response 2/2**
>
> **[Q3] Please include statistics for class distribution - this will provide a better idea of imbalance as well.**
>
> ClothingADC includes 1000 subclasses under each of the twelve-main-class. The number of samples in each subclass is close to a uniform distribution with an average of 89.73(±2.03) samples per subclass. Thus ClothingADC has subclass balance much higher than most other datasets. Because the dataset is large, users of the dataset can create subclass imbalance if their research requires it by simply removing samples from some classes. This is what we did in section 4 class-imbalance learning benchmarks. More details are in Appendix C.5
>
> **[Q4] The following sentence is a bit unclear "For applications where some label noise can be tolerated, existing data curation software capable of identifying and filtering out irrelevant images, such as Docta, CleanLab , and Snorkel 1, etc."**
>
> In the ideal world, we would love to gather a giant clean dataset for free. However, that's not practical. The intent of this sentence was to introduce ways to remove noisy samples from the dataset.
>
> **Why filter:** ADC after step 2 results in a much larger dataset without human labeling efforts compared to TDC. A large prelabeled-set allows ADC to aggressively remove noisy labeled samples while maintaining the resulting dataset scale. The prelabeled-set is also human effort free, thus filtering samples in ADC won’t waste any human labeling efforts, which is required in the traditional setup.
>
> **Auto filter:** For applications where some label noise can be tolerated, we employ label noise detection methods like [1] to significantly reduce errors. These methods can effectively identify and correct noisy labels, resulting in a more accurate dataset [2]. We found a **79.0%** agreement between automated and human-in-the-loop curation methods. Additionally, we observed a reduction in label noise from **22.2%** to **10.7%** after applying the ADC auto-curation, demonstrating its effectiveness in improving dataset quality.
>
> **Human-in-the-loop filter:** For domains requiring clean data, we advocate for human involvement in addition to algorithmic approaches to ensure perfect annotations. Unlike traditional pipelines where humans are asked to label samples from scratch, our ADC pipeline provides a large amount of noisy labeled samples for humans to review and select the accurate ones. This approach is mentally easier (and thus faster) for the human annotator and results in a very clean dataset, since the selected samples have guaranteed human and machine label agreements. Appendix Table 7 shows a detailed cost analysis.
>
> [1] Zhu, Zhaowei, Yiwen Song, and Yang Liu. "Clusterability as an alternative to anchor points when learning with noisy labels." International Conference on Machine Learning. PMLR, 2021.
>
> [2] Zhu, Zhaowei, Jialu Wang, Hao Cheng, and Yang Liu. "Unmasking and Improving Data Credibility: A Study with Datasets for Training Harmless Language Models." In The Twelfth International Conference on Learning Representations, 2024.

---

> > ### Comment · Reviewer_8uBb · 2024-11-22
> >
> > Thanks for your responses.

---

### Meta-Review · Area_Chair_f59t · 2024-12-20

**Metareview:**

This paper was reviewed by four experts in the field and received 8, 5, 5, 6 as the final ratings. The reviewers agreed that the paper tackles an important problem of dataset creation, it is well-written and easy to follow, and the hyper-parameter details are clearly specified. The created Clothing-ADC dataset, with more than 1 million samples and 12k classes, will be useful to the community.

Reviewers YU7i and kh2s raised a concern about the novelty of the method, as it mostly uses existing methods for dataset creation, handling label noise and class imbalance. While the authors have explained in the rebuttal that this may be appropriate for the "datasets and benchmarks" track of the conference, the AC believes that the novelty is still limited, given the standards of ICLR. Moreover, a thorough comparison against other automated dataset creation pipelines is necessary to appropriately understand the usefulness and merit of the proposed ADC methodology.

Reviewer YU7i also mentioned that the connection between the benchmarks (focused on label noise detection and class imbalance learning) and the goal of automated dataset construction is unclear. The authors have explained in the rebuttal that the benchmarks have been designed to validate the effectiveness of the ADC software solutions (the label noise and class imbalance learning benchmarks are used to demonstrate the effectiveness when working with imperfect training data). However, the class-imbalance learning baselines and the label noise detection baselines studied in the paper are all existing methods and none of them have been proposed by the authors. Thus, the need to validate their effectiveness remains unclear and disconnected from the main objective of the paper. A question was also raised about other kinds of data issues such as outliers, duplicates, blurry images, which was not addressed convincingly by the authors in the rebuttal.

A few ethical concerns were also raised, such as the filtering of inappropriate or unethical images and addressing copyright concerns. These were not addressed convincingly by the authors and their reliance on the "safe search" option of search engines is insufficient to safeguard against insufficient or unethical data.

We appreciate the authors' efforts in meticulously responding to each reviewer's comments. However, in light of the above discussions, we conclude that the paper may not be ready for an ICLR publication in its current form. While the paper clearly has merit, the decision is not to recommend acceptance. The authors are encouraged to consider the reviewers' comments when revising the paper for submission elsewhere.

**Additional Comments On Reviewer Discussion:**

Please see my comments above.

---

### Decision · Program_Chairs · 2025-01-22

Reject